# Study to Determine the Prevalence of Menstrual Migraine in Reproductive-Age Women in Saudi Arabia

**DOI:** 10.3390/healthcare12030317

**Published:** 2024-01-25

**Authors:** Zainah Al-Qahtani, Bayapa Reddy Narapureddy, Lingala Kalyan Viswanath Reddy, Hassan Yahya M. Asiri, Ahmed Abdullah H. Alsulami, Nawaf Khalid Ahmed Hassan, Rammas Abdullah Shawkhan, Nouf Abdulraheem Hamood, Hussein Ahmed M. Almahdi, Yousef Yahya Al Qasim, Yahya Ayed Mohammed Al Majbar, Abdullah Ali A. Swadi, Abdulbari Hadi H. Asiri, Bassam Ahmed A. Almaker

**Affiliations:** 1Neurology Section, Internal Medicine Department, College of Medicine, King Khalid University, Abha 61421, Saudi Arabia; zahalas@kku.edu.sa; 2Department of Public Health, College of Applied Medical Sciences, Khamis Mushayt, King Khalid University, Abha 61421, Saudi Arabia; 3Department of Public Health, College of Health Sciences, Saudi Electronic University, Abha 61421, Saudi Arabia; lv.reddy@seu.edu.sa; 4College of Medicine, King Khaled University, Abha 61421, Saudi Arabia; 438801414@kku.edu.sa (H.Y.M.A.); 438801452@kku.edu.sa (A.A.H.A.); 438801448@kku.edu.sa (N.K.A.H.); 438800401@kku.edu.sa (R.A.S.); 438800368@kku.edu.sa (N.A.H.); 438801411@kku.edu.sa (H.A.M.A.); 438801386@kku.edu.sa (Y.Y.A.Q.); 438801383@kku.edu.sa (Y.A.M.A.M.); 439800402@kku.edu.sa (A.A.A.S.); 5College of Pharmacy, King Khalid University, Abha 61421, Saudi Arabia; 442801184@kku.edu.sa (A.H.H.A.); 442801186@kku.edu.sa (B.A.A.A.)

**Keywords:** menstrual migraine, migraine, non-menstrual migraine, premenstrual dysphoric disorder, premenstrual syndrome, reproductive-age women, Saudi Arabia

## Abstract

Background: Migraine is a common health condition in both men and women. Premenstrual syndrome (PMS) affects many women during their menstrual cycle, with around 50–60% of women with migraine attacks experiencing menstrual headaches. Most have mild symptoms, but 5–8% suffer from moderate to severe symptoms, causing distress and functional issues. Pure menstrual migraine (PMM) occurs in about 50% of women with migraine, and it can be debilitating in terms of frequency and severity. This information is crucial for Saudi Arabian medical professionals to provide better care and support, improving the quality of life for women with PMS and menstrual migraine (MM) attacks. Objectives: To estimate the prevalence of MM in women, to evaluate the severity and frequency of MM in women with PMS, and to identify potential risk factors aggravating MM in women with PMS in Saudi Arabia. Methodology: A cross-sectional community-based study was conducted on reproductive-aged (18–50 years) women who had regular menstrual cycles and were diagnosed with PMS, using a self-administered questionnaire between December 2022 to May 2023 in Saudi Arabia. Results: Out of the 2130 female participants, 397 (18.6%) had migraine. Among these 397 migraine sufferers, 230 (57.9%) experienced MM, while 167 (42.1%) had non-MM. In reproductive women in general, MM occurred in 10.7% of cases, while non-MM was observed in 7.8%. There is a correlation between increasing BMI and an increased incidence of MM. About one-third of the participants experienced moderate disability due to migraine attacks, with 134 (33.8%) individuals affected. Additionally, most MM sufferers missed at least 3 days of work in the last 3 months due to their condition. Conclusions: Migraine attacks occurring during the menstrual cycle impair the ability to engage in social, physical, household, and academic activities, often hindering the fulfillment of professional commitments. To gain a deeper understanding of menstrual and non-menstrual migraine attacks, it is essential to conduct extensive prospective studies aimed at developing effective management strategies.

## 1. Introduction

A migraine headache is characterized by intense throbbing or pulsing pain, often on one side of the head. It can be accompanied by nausea, vomiting, and sensitivity to light and sound. The overall global prevalence of migraine is approximately 14–15% [1]. Globally, migraine prevalence is doubled in women, at 17–21%, compared to men, at 6–10% [2,3]. After menarche, women experience three times more migraine attacks than men, constituting 70% of cases globally [4]. Among women with migraine, 50–60% also suffer from menstrual headaches during premenstrual syndrome (PMS) [5]. In nearly 5–8% of women with PMS, migraine severely impacts daily activities and may last for hours to days, hampering daily life and well-being [6,7,8].

A menstrual migraine (MM) is a type of migraine headache that occurs during a woman’s menstrual cycle. These migraine attacks often coincide with the hormonal fluctuations that happen before, during, or after menstruation. Menstrual migraine attacks, with aura or without aura, classified into menstrual-related migraine (MRM), pure menstrual migraine (PMM), and non-menstrual migraine (NMM), occur 2 days before or during the first 3 days of menstruation, disrupting daily activities [9,10,11]. Pure menstrual migraine (PMM) coincides with the menstrual cycle in 50% of women with migraine attacks, often being frequent and severe [12,13]. In PMM, about 70% of women experience premenstrual pain, with almost half reporting migraine attacks on the first day of their menstrual cycle [14]. Most women with migraine note a link to their menstrual cycles, experiencing true menstrual migraine only on the first or second day of menstruation [15]. Menstrual-related migraine attacks can occur at any point during menstruation, including the first 2 days and other cycle phases [16]. Managing PMM may involve lifestyle adjustments, medication, or hormonal treatments, and consulting with a healthcare professional is recommended for personalized advice.

Genetic factors, family history, and hormonal fluctuations contribute to PMM, and PMS exacerbates menstrual migraine attacks [17,18,19,20]. Collaborating with healthcare providers for personalized treatment plans is crucial for women experiencing pure menstrual migraine [21]. Premenstrual dysphoric disorder (PMDD), an intense form of PMS, involves physical and emotional symptoms associated with hormonal fluctuations [22].

A study correlating PMS and PMM found that 77% of women with PMM experienced PMS symptoms [20]. Approximately 60% of women with migraine experience attacks associated with their menstrual cycle [20]. In the Saudi Arabian context, the available data suggest a critical need for understanding and addressing the issue of migraine, particularly menstrual migraine. Limited data in Saudi Arabia suggest that 50% of women with migraine experience menstrual migraine attacks, with a significant association between PMS and menstrual migraine [23]. Lifestyle adjustments and managing PMS symptoms can alleviate migraine attacks, improving the quality of life for affected women [24]. This underscores the importance for healthcare providers in Saudi Arabia to be aware of these connections for enhanced medical care and support.

The dearth of extensive research on menstrual migraine in Saudi Arabia points to a significant gap in knowledge. While general migraine studies exist, there is a lack of detailed exploration into the unique aspects of MM in this cultural and geographical context. The proposed study seeks to fill this void by providing a thorough investigation into the prevalence, associated factors, and impact of menstrual migraine among reproductive-age women in Saudi Arabia. Addressing this knowledge gap is crucial not only for advancing our understanding of migraine disorders but also for tailoring effective healthcare interventions specific to the Saudi Arabian population.

## 2. Objectives

To estimate the prevalence of PMM in reproductive-age women in Saudi Arabia, to evaluate the severity and frequency of PMM in women with PMDD in Saudi Arabia, and to identify potential factors aggravating the development of PMM in women with PMDD in Saudi Arabia.

## 3. Methodology

Study Design: A cross-sectional study was employed with a self-administered questionnaire, spanning from December 2022 to May 2023 in Saudi Arabia (Please see Appendix A).

Study Population: Literate women aged between 18 to 50 years, with regular menstrual cycles (cycles typically 21 to 35 days, with menstruation lasting 2 to 7 days), diagnosed with PMS, and living in Saudi Arabia were included in the study. Participants were recruited from various regions of Saudi Arabia within the general population.



### 3.1. Methods

The validated questionnaire was adapted from previous studies and translated into Arabic by local scholars [25,26]. The adopted study tool was modified as per the cultural acceptance and piloted on 28 female medical students. After translating the questionnaire, Cronbach’s alpha was determined as 0.8 and tested the content validity and cultural acceptance of the people. The survey questionnaire encompassed inquiries related to demographics like age, education, living area, marital status, occupation, BMI, family history, anthropometric profiles, lifestyle habits, aggravating factors, symptoms, and the severity of PMS and menstrual migraine. It also included the MIDAS score, Headache Impact Test-6 (HIT-6), and details of prophylactic treatment. A purposive sampling technique was used to recruit the study participants from the general public. Data collection took place through a self-administered online Google Form shared via WhatsApp and Snapchat with selected individual women and women’s groups. After consenting to participate, each participant was requested to complete the self-administered questionnaire, including their demographic and anthropometric data, menstrual cycles, and the impact of migraine headaches.

Diagnostic Criteria According to International Classification of Headache Disorders, 3rd Edition (ICHD-3).

### 3.2. Pure Menstrual Migraine

Migraine attacks in a reproductive-age woman meeting criteria without aura.Migraine headache attacks occur during menstruation or 2 days before menstruation, lasting until the third day of menstruation.
-Attacks occur in at least 2 out of 3 menstrual cycles.-Do not occur at other times during the menstrual cycle.

### 3.3. Menstrual-Related Migraine

-Migraine attacks in a reproductive-age woman fulfilling criteria without aura.-Attacks happen on menstruation days or 2 days before, continuing until the third day of menstruation.
-Occur in at least 2 out of 3 menstrual cycles.-Can also occur at any time during the menstrual cycle.

The diagnosis of menstrual migraine relied on self-reported history rather than confirmation through headache and menstruation diaries.

Sampling: Participants were selected using a purposive nonprobability sampling technique.

### 3.4. Inclusion and Exclusion Criteria

Women aged between 18 years and 50 years, with regular menstruation and living in Saudi Arabia who provided digital consent to participate were included. While the sample may not have been entirely representative of the population of women with migraine, it offered valuable insights into PMM with PMS. Exclusions comprised women under 18 and over 50 years of age, those severely ill, currently on hormonal contraceptive therapy, with severe neurological problems (stroke, epilepsy, brain tumors, and psychiatric treatment history), illiterate individuals, those diagnosed with polycystic ovarian syndrome, women who had undergone hysterectomy or oophorectomy, those with severe depression, and hospitalized individuals.

### 3.5. Ethical Issues

The study authors provided an explanation of the study’s objectives and a brief description in the local Arabic language before participants filled out the questionnaire. Following the Declaration of Helsinki, digital consent was obtained from participants before commencing the study. Institutional ethical approval was acquired from the Research Ethics Committee (HAPO-06-B-001) of King Khalid University, Abha, KSA, Institutional Ethical approval (ECM#2022-2505), with an approval duration from 15 September 2022 to 14 September 2023.

### 3.6. Statistical Analysis

The validated data underwent analysis using the Statistical Package for the Social Sciences (IBM SPSS Statistics) for Windows, version 21 (IBM Corp., Armonk, NY, USA). All collected data were exported to a Microsoft Office 2019 Excel spreadsheet from Google Forms. Categorical variables, including demographic data, educational data, PMM, PMS, clinical features, and aggravating factors, were expressed in proportions and inter-quartile range, and for hypothesis testing, the Chi-square test and other relevant tests were used. A *p*-value of <0.05 was considered statistically significant at 80% power and 95% confidence interval level.

## 4. Results

This study aimed to investigate the prevalence of menstrual migraine attacks in women experiencing PMS. Out of the 2130 participants, the majority, 1733 (81.4%), had PMDD without migraine, nearly 301 (14.1%) met the ICHD-3 Criteria for migraineurs with premenstrual symptoms, and a minimal 96 (4.5%) had migraine without PMDD. The average age of participants was 24 years, with an interquartile range of 21–35, indicating that PMS with or without migraine tends to occur at a younger age. Among the 301 participants with PMM and PMS, over half were overweight or obese. Within this group, 164 (54.5%) were single, 221 (73.4%) had graduated or completed higher education, and 271 (90%) were nonsmokers. Among the 96 participants with non-MM, nearly three-fourths, 68 (70.8%), had a family history of headaches (Figure 1). In the group of 301 participants with migraine and PMDD, the majority, 189 (62.8%), also had a family history of headaches. Approximately 10% of participants with PMDD-associated migraine were smokers (Table 1).

Among individuals with migraine attacks and PMDD, more than half, specifically 58.5%, experienced PMM. Similarly, among those with migraine but without PMDD, an almost equal proportion had menstrual migraine. Out of the 230 individuals with menstrual migraine, 159 (40.0%) reported experiencing migraine attacks 2 days before menstruation, and approximately one-fifth (17.89%) experienced headaches during the first 3 days of menstruation. The majority of PMDD sufferers experienced unilateral localization (318, 80.1%), pulsatile pain (301, 75.8%), and nausea (269, 67.8%). Among participants with migraine, with or without PMDD, the factors that aggravated their condition included sleep disturbance (325, 81.9%), menstruation (260, 65.5%), prolonged fasting (216, 54.4%), and more (Table 2) & Figure 2.

Out of the 2130 female participants, 397 (18.6%) suffered from migraine attacks. Among these 397 migraine sufferers, 230 (57.9%) had pure menstrual migraine, while 167 (42.1%) experienced non-MM. The prevalence of pure menstrual migraine was 10.8%, and of non-menstrual migraine, 7.8%. PMM is more common among middle-aged women, with an average age of 33 years and an interquartile range of 24–40 years. Non-MM individuals tended to be younger, with an average age of 24 years and an interquartile range of 21–34 years. There was a correlation between increasing BMI and an increased incidence of menstrual migraine attacks.

Among single individuals, non-MM (101, 60.5%) was more common than PMM (98, 42.6%). Education appeared to be a contributing factor, with more participants with higher education suffering from migraine attacks. Among migraineurs, symptoms were frequently aggravated by movement in 277 (69.8%) cases. Photophobia was the most common symptom (342, 86.1%), followed by unilateral localized pain (318, 80.1%). Among the factors that aggravated migraine attacks, sleep disturbance was the most common (325, 81.9%), followed by menstruation (260, 65.5%), and prolonged fasting (216, 54.4%). In terms of migraine disability, around one-third of participants suffered from moderate disability (134, 33.8%).

The frequency of headache attacks was significantly higher in PMM sufferers (median 8 episodes, interquartile range 5–12) compared to other types of migraineurs, and the difference was statistically significant (*p* < 0.001). The Headache Impact Test-6 (HIT-6) assessed headaches’ impact on daily life, with a severity score of 66 (IQR 61–69), consistent across all migraine groups. The MIDAS score, evaluating migraine frequency and impact, showed a higher median (66, IQR 63–69) in PMM sufferers, though the difference was not statistically significant (*p* > 0.05). About one-third of PMM individuals experienced severe to moderate disability due to migraine (Table 3), (Figure 2).

Among the 397 migraineurs, 301 (76%) experienced migraine with PMDD, while one-quarter of them 96 (24%) did not have PMDD. Approximately three-fourths of these migraineurs 288 (72.5%) suffered from a very severe form of headache during their migraine episodes. Nearly half of them 199 (50.1%) had severe headaches very often, and one-quarter 89 (22.4%) always experienced severe headaches during episodes.

The majority, 228 (57.5%) reported that their headaches often or always limited their ability to perform their usual daily activities. Most participants 347 (87.8%) preferred to lie down when they experienced a headache. Over the past four weeks, approximately 169 (45%) frequently felt tired due to their headaches, and a significant number 193 (48.7%) commonly only experienced irritable moods because of their headaches (Table 4).

The severity of migraine was assessed with the MIDAS score. Among the 397 migraineurs, 167 (42%) experienced NMM, while 230 (58%) had PMM attacks. A significant proportion of MM sufferers were absent from their duties for at least 3 days in the last 3 months due to their menstrual migraine attacks. These individuals also reported higher rates of absence from social activities and family functions (Table 5).

## 5. Discussion

In this comprehensive study, it was noted that females with menstrual migraine (MM) exhibited a higher prevalence of premenstrual symptoms (PMS) compared to those without MM. Interestingly, while HIT-6 scores and migraine severity did not show significant differences, the duration of migraine attacks was comparable between the two groups. However, self-reported MM sufferers demonstrated higher MIDAS scores in contrast to non-menstrual migraineurs.

Among the 2130 participants, a significant portion (81.4%) experienced premenstrual dysphoric disorder (PMDD) without migraine, whereas 14.1% reported migraine with premenstrual symptoms, and a minor fraction (4.5%) had migraine without PMDD. The average age of participants was 28 years, with variations noted in the average age between premenstrual migraineurs (33 years) and non-MM sufferers (24 years). These findings align with studies by Chalmer MA et al. [11] and Vetvik KG et al. [27] which observed age-related patterns in women with MM, whereas Tschudin et al. [28] observed peak PMS among women aged 35 years.

Examining the prevalence among participants, it was found that 18.6% experienced migraine. This aligns with similar studies by Witteveen H et al. [29] and Muayqil T et al. [30] highlighting the consistency of migraine prevalence across different populations. Notably, studies on the Saudi population by Bamalan BA et al. [31] Balaha M et al. [24], and Hanadi Bakhsh et al. [23] reported varying prevalence rates of menstrual cycle-related migraine headaches and PMS. Further analysis revealed that among the migraineurs, 14.1% suffered from PMM, with a higher prevalence compared to non-MM (4.5%). These findings resonate with studies conducted by Yamada K et al. [32] and Wang SJ et al. on the Taiwanese population, emphasizing the association between PMS and migraine [21].

Education levels played a role, with most PMM sufferers being graduates (68.4%), aligning with observations by Pavlovic JM et al. [33]. The study also highlighted a correlation between body mass index (BMI) and PMM attacks, noting an increased prevalence with higher BMI, in line with studies by Peterlin BL et al. [34] and Pavlović JM et al. [33]. Interestingly, family history emerged as a significant factor, positively associated with both menstrual migraine (62.8%) and non-menstrual migraine (70.8%). This finding supports the notion that genetic factors play a role in migraine susceptibility, as noted by Russell MB et al. [35].

The temporal aspect of PMM attacks was explored, revealing that most menstrual migraineurs reported headaches starting 2 days before menstruation, with a notable proportion (17.8%) experiencing continued headaches during the first 3 days of menstruation. These findings coincide with earlier studies linking migraine and PMM with premenstrual symptoms [36,37]. The impact of PMM on daily life was evident, with participants reporting moderate to severe pain, nausea, photophobia, and disruptions in routine activities. This aligns with observations by Balah M et al. [24] on the Saudi population, indicating a significant impact on concentration and academic performance.

Women with PMM in this study reported nearly 8 attacks (with a range of 5–12) this year associated with the menstrual cycle. Around 73% of them reported moderate to severe pain, 68% experienced nausea, and 86.1% reported photophobia. Migraine had a higher impact in terms of elevated HIT scores (66 with a range of 63–69), more migraine days, and longer attack durations. PMS symptoms were similarly common in women with and without MM. Triggers for PMM attacks were varied, with sleep disturbances (86.4%), menstruation (70%), and movements (67.7%) being prominent factors. These findings are consistent with studies by Bamalan BA et al. [31] and Moy G et al. [16], highlighting the multifaceted nature of migraine triggers. Witteveen H et al. [29] noted that the PMM attacks were aggravated by physical activity at 63.6% among menstrual migraineurs.

Very few participants were availing themselves of prophylactic medications for migraine. A study by Mehkri Y et al. [38] mentioned that Calcitonin Gene-Related Peptide (CGRP), in modulating physiological processes of migraine antagonists, had a great effect on the treatment of migraine, along with lifestyle changes and stress reduction, and consulting with a healthcare professional for personalized advice on headaches. Sorrentino ZA et al. [39] mentioned multi-modal pain control strategies, including nerve blocks and preventive measures like regular exercise and a balanced diet. Remarkably, a small percentage (5.2%) of women reported using antipsychotic drugs as migraine prophylaxis. However, the study did not delve into possible comorbid psychiatric illnesses, emphasizing the need for further exploration in this area.

This extensive study sheds light on the complex interplay of factors influencing menstrual migraine, encompassing demographics, education, BMI, family history, temporal patterns, and triggers. The findings contribute valuable insights for understanding and managing this specific subtype of migraine, paving the way for more targeted interventions and treatment approaches.

## 6. Limitations

The main limitation of this study is reliance on self-reported data rather than prospective diaries, especially regarding migraine symptoms, PMDD, and lifestyle factors, which introduces the possibility of recall bias and subjective interpretation. Participants may not accurately recall the timing and severity of symptoms, impacting the precision of the reported associations. The study predominantly focused on a specific demographic, and the findings may not apply to diverse populations with different cultural, socioeconomic, or geographic characteristics. The study identified several triggers for menstrual migraine attacks, but it did not delve into the nuances of individual variations in trigger responses. Factors such as stress levels, coping mechanisms, and lifestyle habits were not thoroughly examined, potentially overlooking relevant contributors. This study followed the ICD-10 criteria for the diagnosis of PMM migraine attacks happening more frequently 2 days before or during menses, without considering the rate of occurrences with menstruation.

## 7. Conclusions and Recommendations

This community-based study of Saudi reproductive-age women is representative of the Saudi Arabian population. This study’s results are purely based on self-reported migraine attacks among reproductive-age women and their association with the menstrual cycle. These migraine attacks during the menstrual cycle impaired their social and physical ability to perform their routine household activities and academic activities and rendered them unable to fulfill their professional commitments. To better understand MM and non-MM, large prospective studies are required to plan the best management methods for MM and non-MM cases and, as a result, reduce the burden of migraine-related physical and psychological health problems.

## Figures and Tables

**Figure 1 healthcare-12-00317-f001:**
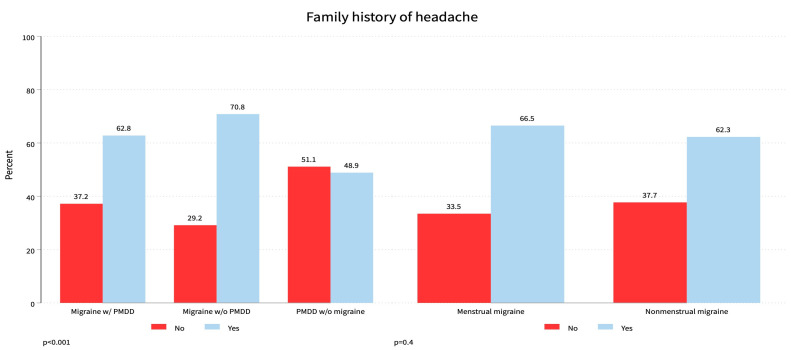
Family history of headache and that associated with menstrual migraine.

**Figure 2 healthcare-12-00317-f002:**
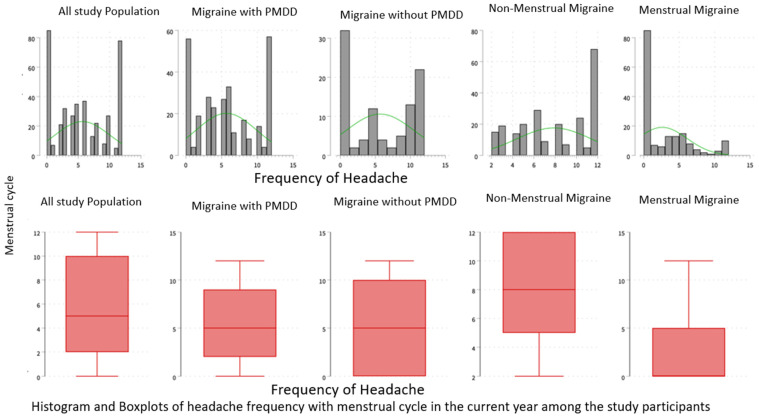
Migraine headache frequency associated with the menstrual cycle.

**Table 1 healthcare-12-00317-t001:** Demographic Characteristics of the Study Population with PMDD.

		Total	Migraine with PMDD	Migraine without PMDD	PMDD without Migraine	
Characteristics		n = 2130	n = 301 (14.1)	n = 96 (4.5)	n = 1733 (81.4)	*p* Value
Prevalence		-	14.1%	4.5%	81.3%	
Age, median (IQR), years		24 (21–35)	27 (22–39)	35 (23–40)	24 (21–33)	<0.001 *
Weight, median (IQR), kg		60 (51–70)	60 (51.5–70)	62 (52–74)	59 (51–69)	0.1 ^#^
Height, median (IQR), m		1.6 (1.5–1.6)	1.6 (1.5–1.6)	1.6 (1.5–1.6)	1.6 (1.5–1.6)	0.04 *
BMI, median (IQR), kg/m^2^		23.3 (20.4–27.2)	24.2 (20.7–27.5)	24.9 (21.5–29.5)	23.1 (20.4–27.1)	0.004 *
BMI, No. (%)	Underweight (<18.5)	235 (11)	27 (9)	8 (8.3)	200 (11.5)	0.003 *
	Normal weight (18.5–24.9)	1086 (51)	139 (46.2)	40 (41.7)	907 (52.3)	
	Overweight (25–29.9)	472 (22.9)	89 (29.6)	25 (26)	358 (20.7)	
	Obese (≥30)	319 (14.9)	46 (15)	23 (24)	250 (14.4)	
Obesity	Class–I (30–34.9)	224(10.5)	23 (51.1)	17 (73.7)	184 (73.6)	
	Class–II (35–39.9)	61 (2.9)	14 (31.1)	4 (17.4)	43 (17.2)	
	Class–III (≥40)	34 (1.5)	9 (17.8)	2 (8.7)	23 (9.2)	
Saudi Arabia region, No. (%)	Central	399 (18.7)	49 (16.3)	16 (16.7)	334 (19.3)	0.001 *
	Eastern	431 (20.2)	81 (26.9)	32 (33.3)	318 (18.3)	
	Northern	114 (5.4)	20 (6.6)	3 (3.1)	91 (5.3)	
	Southern	704 (33.1)	84 (27.9)	29 (30.2)	591 (34.1)	
	Western	482 (22.6)	67 (22.3)	16 (16.7)	399 (23)	
Marital status, No. (%)	Divorced	60 (2.8)	6 (2)	4 (4.2)	50 (2.9)	0.001 *
	Married	773 (36.3)	128 (42.5)	56 (58.3)	589 (34)	
	Single	1289 (60.5)	164 (54.5)	35 (36.5)	1090 (62.9)	
	Widowed	8 (0.4)	3 (1)	1 (1)	4 (0.2)	
Education level, No. (%)	Primary	1 (0)	—	—	1 (0.1)	0.2 ^#^
	Intermediate	17 (0.8)	2 (0.7)	2 (2.1)	13 (0.8)	
	High school	424 (19.9)	51 (16.9)	20 (20.8)	353 (20.4)	
	Diploma	114 (5.4)	27 (9)	6 (6.2)	81 (4.7)	
	Graduate	1472 (69.1)	206 (68.4)	64 (66.7)	1202 (69.4)	
	Postgraduate	102 (4.8)	15 (5)	4 (4.2)	38 (4.8)	
Pregnancy	No	2051 (96.3)	290 (96.3)	95 (99)	1666 (96.1)	0.4
	Yes	79 (3.7)	11 (3.7)	1 (1)	67 (3.9)	
Smoking	No	2002 (94)	271 (90)	92 (95.8)	1639 (94.6)	0.01 *
	Yes	128 (6)	30 (10)	4 (4.2)	94 (5.4)	
Family history of headache	No	1026 (48.2)	112 (37.2)	28 (29.2)	886 (51.1)	0.001 *
	Yes	1104 (51.8)	189 (62.8)	68 (70.8)	847 (48.9)	

* Significant, ^#^ Not Significant.

**Table 2 healthcare-12-00317-t002:** Distribution of PMS and Migraine-associated Factors.

		Total	Migraine with PMDD	Migraine without PMDD	PMDD without Migraine	
Characteristics		n = 2130	n = 301 (14.1)	n = 96 (4.5)	n = 1733 (81.4)	*p* Value
MM n = 397	No	167 (42.1)	125 (41.5)	42 (43.8)	NA	0.7 ^#^
	Yes	230 (57.9)	176 (58.5)	54 (56.2)	NA	
Time of headache during menstruation n = 397	Two days before menstruation	159 (40.0%)	123 (40.8)	36 (37.5%)	NA	0.6 ^#^
	The first three days of menstruation	71 (17.8%)	53 (17.6%)	18 (18.7%)	NA	
	Mid of the menstruation	167 (42.2%)	125 (41.6%)	42 (43.8%)	NA	
Headache attack frequency with menses in the current year, median (IQR)		5 (2–10)	5 (2–10)	5 (0–10)	NA	1 ^#^
Migraine symptoms	Aggravated by movement	269 (67.7)	206 (68.4)	63 (65.6)	NA	0.01 *
	Moderate–severe pain	277 (69.7)	221 (73.4)	56 (58.3)	NA	0.03 *
	Nausea without vomiting	79 (19.8)	63 (20.9)	16 (16.7)	NA	0.6 ^#^
	Photophobia	44 (11.1)	35 (11.6)	9 (9.4)	NA	0.6 ^#^
	Prolonged symptoms (≥72 h)	397 (100)	301 (100)	96 (100)	NA	NA
	Pulsatile pain	270 (68.0)	228 (75.7)	62 (64.6)	NA	0.3 ^#^
	Unilateral localization	318 (80.1)	241 (80.1)	77 (80.2)	NA	1 ^#^
	Vomiting	79 (19.9)	32 (10.6)	47 (48.9)	NA	0.5 ^#^
Aggravating factors	Foods	44 (11.1)	35 (11.6)	9 (9.4)	NA	0.7 ^#^
	Menstruation	260 (65.5)	210 (69.8)	50 (52.1)	NA	0.002 *
	Prolonged fasting	216 (54.4)	181 (60.1)	35 (36.5)	NA	0.001 *
	Sleep disturbances	325 (81.9)	260 (86.4)	65 (67.7)	NA	0.001 *
	Smells	1 (0.3)	1 (0.3)	—	NA	1 ^#^
	Sounds	1 (0.3)	—	1 (1)	NA	0.2 ^#^
	Stress	3 (0.8)	1 (0.3)	2 (2.1)	NA	0.1 ^#^
	Visual	2 (0.5)	—	2 (2.1)	NA	0.1 ^#^
HIT-6 score, median (IQR)		66 (62–68)	66 (63–69)	65 (61–68)	NA	0.3 ^#^
Migraine severity	Minimal (<50)	3 (0.8)	1 (0.3)	2 (2.1)	NA	0.1 ^#^
	Mild (50–55)	14 (3.5)	8 (2.7)	6 (6.2)	NA	
	Moderate (56–59)	38 (9.6)	28 (9.3)	10 (10.4)	NA	
	Severe (≥60)	342 (86.1)	264 (87.7)	78 (81.2)	NA	
MIDAS score, median (IQR)		14 (9–24)	16 (9–27)	11.5 (7–20)	NA	0.02 *
Migraine disability	I–Little or no disability	55 (13.9)	35 (11.6)	20 (20.8)	NA	0.05 *
	II–Mild disability	84 (21.2)	60 (19.9)	24 (25)	NA	
	III–Moderate disability	134 (33.8)	105 (34.9)	29 (30.2)	NA	
	IV–Severe disability	124 (31.2)	101 (33.6)	23 (24)	NA	
Prophylaxis therapy	No	368 (96.8)	278 (95.9)	90 (100)	NA	0.1 ^#^
	Yes	12 (3.2)	12 (4.1)	—	NA	
Prophylaxis medication	Amitriptyline	4 (33.3)	4 (33.3)	—	NA	
	Propranolol	2 (16.7)	2 (16.7)	—	NA	
	Topiramate	4 (33.3)	4 (33.3)	—	NA	
	Valproic acid	2 (16.7)	2 (16.7)	—	NA	
PMDD symptoms	Anxiety	1642 (77.1)	254 (84.6)	NA	1363 (78.6)	0.02 *
	Apathy and loss of interest	1117 (52.4)	188 (62.5)	NA	918 (53)	0.003 *
	Appetite changes	1280 (60.1)	202 (67.1)	NA	1053 (60.8)	0.04 *
	Confusion	690 (32.4)	124 (41.2)	NA	563 (32.5)	0.004 *
	Depressed mood	1510 (70.9)	246 (81.7)	NA	1246 (71.9)	0.001 *
	Easily irritable	1710 (80.3)	257 (85.4)	NA	1414 (81.6)	0.1 ^#^
	Fatigue	1773 (83.2)	273 (90.7)	NA	1467 (84.7)	0.01 *
	Loss of concentration	875 (41.1)	170 (56.5)	NA	700 (40.4)	0.001 *
	Mood disturbances	1905 (89.4)	276 (91.7)	NA	1577 (91)	0.7 ^#^
	Sleep disturbances	1152 (54.1)	188 (62.5)	NA	948 (54.7)	0.01 *
	Somatic symptoms	1632 (76.6)	252 (83.7)	NA	1333 (76.9)	0.01 *
BMI: Body-mass index; HIT-6: Headache Impact Test; MIDAS: Migraine Disability Assessment questionnaire; NA: Not applicable; PMDD: Premenstrual dysphoric disorder.						

* Significant, ^#^ Not Significant.

**Table 3 healthcare-12-00317-t003:** Distribution of Migraineur Participants by Demographics with Menstrual Migraine.

	Total	Pure Menstrual Migraine	Non-Menstrual Migraine	
Characteristics	n = 397	n = 230 (57.9)	n = 167 (42.1)	*p* Value
Age, {IQR} (years)	28 (22–40)	33 (24–40)	24 (21–35)	<0.001 *
Weight, (IQR), kg	60 (52–70)	62 (54–72)	58.5 (50–70)	0.01 *
Height, (IQR), m	1.6 (1.5–1.6)	1.6 (1.5–1.6)	1.6 (1.5–1.6)	0.4 ^#^
BMI, (IQR), kg/m^2^	24.3 (21.1–27.6)	25.2 (21.6–28.7)	23.2 (20.1–27)	0.01 *
BMI, No. (%)				<0.001 *
Underweight (<18.5)	35 (8.8)	12 (5.9)	23 (13.9)	
Normal weight (18.5–24.9)	179 (45.2)	96 (41.7)	83 (50)	
Overweight (25–29.9)	114 (28.8)	81 (35.2)	33 (19.9)	
Obese (≥30)	68 (17.2)	41 (17.8)	27 (16.3)	
Obesity, No. (%)				0.047 *
Class–I (30–34.9)	40 (58.8)	24 (58.5)	16 (59.3)	
Class–II (35–39.9)	18 (26.5)	14 (34.1)	4 (14.8)	
Class–III (≥40)	10 (14.7)	3 (7.3)	7 (25.9)	
Saudi Arabia region, No. (%)				0.1 ^#^
Central	65 (16.4)	35 (15.2)	30 (18)	
Eastern	113 (28.5)	76 (33)	37 (22.2)	
Northern	23 (5.8)	16 (7)	7 (4.2)	
Southern	113 (28.5)	56 (24.3)	57 (34.1)	
Western	83 (20.9)	47 (20.4)	36 (21.6)	
Marital status, No. (%)				0.002 *
Single	199 (50.1)	98 (42.6)	101 (60.5)	
Married	184 (46.3)	124 (53.9)	60 (35.9)	
Divorced	10 (2.5)	5 (2.2)	5 (3)	
Widowed	4 (1)	3 (1.3)	1 (0.6)	
Education level, No. (%)				0.3 ^#^
Primary	—	—	—	
Intermediate	4 (1)	3 (1.3)	1 (0.6)	
High school	71 (17.9)	38 (16.5)	33 (19.8)	
Diploma	33 (8.3)	19 (8.3)	14 (8.4)	
Graduate	270 (68)	155 (67.4)	115 (68.9)	
Postgraduate	19 (4.8)	15 (6.5)	4 (2.4)	
Pregnant, No. (%)				0.6 ^#^
No	385 (97)	224 (97.4)	161 (96.4)	
Yes	12 (3)	6 (2.6)	6 (3.6)	
Smoking, No. (%)				1 ^#^
No	363 (91.4)	210 (91.3)	153 (91.6)	
Yes	34 (8.6)	20 (8.7)	14 (8.4)	
Family history of headache, No. (%)				0.4 ^#^
No	140 (35.3)	77 (33.5)	63 (37.7)	
Yes	257 (64.7)	153 (66.5)	104 (62.3)	
Time of headache during menses, No (%)				<0.001 *
Three days before menses	155 (58.7)	147 (63.9)	8 (23.5)	
The first two days of menses	70 (26.5)	64 (27.8)	6 (17.6)	
Middle of menses	19 (7.2)	18 (7.8)	1 (2.9)	
After menses	20 (7.6)	1 (0.4)	19 (55.9)	
Headache attacks frequently with menses in the current year, median (IQR)	5 (2–10)	8 (5–12)	5 (0–5)	<0.001 *
Migraine symptoms. No (%)				
Aggravated by movement	277 (69.8)	158 (68.7)	119 (71.3)	0.7 ^#^
Moderate–severe pain	290 (73)	167 (72.6)	123 (73.7)	0.9 ^#^
Nausea without vomiting	269 (67.8)	160 (69.6)	109 (65.3)	0.4 ^#^
Photophobia	342 (86.1)	200 (87)	142 (85)	0.7 ^#^
Prolonged symptoms (≥72 h)	397 (100)	230 (100)	167 (100)	NA
Pulsatile pain	301 (75.8)	184 (80)	117 (70.1)	0.02 *
Unilateral localization	318 (80.1)	187 (81.3)	131 (78.4)	0.5 ^#^
Vomiting	79 (19.9)	45 (19.6)	34 (20.4)	0.9 ^#^
Aggravating factors, No. (%)				
Foods	44 (11.1)	26 (11.3)	18 (10.8)	1 ^#^
Menstruation	260 (65.5)	215 (93.5)	45 (26.9)	<0.001 *
Prolonged fasting	216 (54.4)	128 (55.7)	88 (52.7)	0.6 ^#^
Sleep disturbances	325 (81.9)	189 (82.2)	136 (81.4)	0.9 ^#^
Smells	1 (0.3)	—	1 (0.6)	0.4
Sounds	1 (0.3)	—	1 (0.6)	0.4 ^#^
Stress	3 (0.8)	—	3 (1.8)	0.1 ^#^
Visual	2 (0.5)	—	2 (1.2)	0.2 ^#^
HIT-6 score, median (IQR)	66 (62–68)	66 (63–69)	65 (61–68)	0.2 ^#^
Migraine severity, No (%)				0.02 *
Minimal (<50)	3 (0.8)	—	3 (1.8)	
Mild (50–55)	14 (3.5)	5 (2.2)	9 (5.4)	
Moderate (56–59)	38 (9.6)	18 (7.8)	20 (12)	
Severe (≥60)	342 (86.1)	207 (90)	135 (80.8)	
MIDAS score, median (IQR)	14 (9–24)	15 (9–27)	13 (6–23)	0.3 ^#^
Migraine disability, No. (%)				<0.001 *
I–Little or no disability	55 (13.9)	17 (7.4)	38 (22.8)	
II–Mild disability	84 (21.2)	58 (25.2)	26 (15.6)	
III–Moderate disability	134 (33.8)	78 (33.9)	56 (33.5)	
IV–Severe disability	124 (31.2)	77 (33.5)	47 (28.1)	
Prophylaxis therapy, No. (%)				0.8 ^#^
No	368 (96.8)	214 (96.4)	154 (97.5)	
Yes	12 (3.2)	8 (3.6)	4 (2.5)	
Prophylaxis medications, No. (%)				0.05 *
Amitriptyline	4 (33.3)	2 (25)	2 (50)	
Propranolol	2 (16.7)	2 (25)	—	
Topiramate	4 (33.3)	4 (50)	—	
Valproic acid	2 (16.7)	—	2 (50)	
BMI: Body-mass index; HIT-6: Headache Impact Test-6 questionnaire; MIDAS: Migraine Disability Assessment questionnaire.				

* Significant, ^#^ Not Significant.

**Table 4 healthcare-12-00317-t004:** Headache Impact Test-6 (HIT-6™) with Migraine and PMDD.

	Total	Migraine with PMDD	Migraine without PMDD	
Questions, No. (%)	n = 397	n = 301 (75.8)	n = 96 (24.2)	*p* Value
1. When you have headaches, how often is the pain severe?				0.1 ^#^
Never	—	—	—	
Rarely	11 (2.8)	6 (2)	5 (5.2)	
Sometimes	98 (24.7)	80 (26.6)	18 (18.8)	
Very often	199 (50.1)	146 (48.5)	53 (55.2)	
Always	89 (22.4)	69 (22.9)	20 (20.8)	
2. How often do headaches limit your ability to do usual daily activities including household work, work, school, or social activities?				0.08 ^#^
Never	2 (0.5)	2 (0.7)	—	
Rarely	30 (7.6)	17 (5.6)	13 (13.5)	
Sometimes	137 (34.5)	111 (36.9)	26 (27.1)	
Very often	165 (41.6)	124 (41.2)	41 (42.7)	
Always	63 (15.9)	47 (15.6)	16 (16.7)	
3. When you have a headache, how often do you wish you could lie down?				0.4 ^#^
Never	2 (0.5)	2 (0.7)	—	
Rarely	4 (1)	2 (0.7)	2 (2.1)	
Sometimes	44 (11.1)	30 (10)	14 (14.6)	
Very often	131 (33)	102 (33.9)	29 (30.2)	
Always	216 (54.4)	165 (54.8)	51 (53.1)	
4. In the past 4 weeks, how often have you felt too tired to do work or daily activities because of your headaches?				0.3 ^#^
Never	6 (1.5)	3 (1)	3 (3.1)	
Rarely	36 (9.1)	26 (8.6)	10 (10.4)	
Sometimes	186 (46.9)	137 (45.5)	49 (51)	
Very often	131 (33)	103 (34.2)	28 (29.2)	
Always	38 (9.6)	32 (10.6)	6 (6.2)	
5. In the past 4 weeks, how often have you felt fed up or irritated because of your headaches?				0.3 ^#^
Never	5 (1.3)	2 (0.7)	3 (3.1)	
Rarely	38 (9.6)	26 (8.6)	12 (12.5)	
Sometimes	161 (40.6)	115 (38.2)	46 (47.9)	
Very often	134 (33.8)	109 (36.2)	25 (26)	
Always	59 (14.9)	49 (16.3)	10 (10.4)	
6. In the past 4 weeks, how often did headaches limit your ability to concentrate on work or daily activities?				0.1 ^#^
Never	8 (2)	3 (1)	8 (4.8)	
Rarely	27 (6.8)	19 (6.3)	11 (6.6)	
Sometimes	163 (41.1)	122 (40.5)	79 (47.3)	
Very often	153 (38.5)	120 (39.9)	50 (29.9)	
Always	46 (11.6)	37 (12.3)	19 (11.4)	

^#^ Not Significant.

**Table 5 healthcare-12-00317-t005:** Distribution of Migraine Disability Assessment (MIDAS) with Menstrual Migraine.

	Total	MM	NMM	
Questions, Median (IQR)	n = 397	n = 230 (57.9)	n = 167 (42.1)	*p* Value
On how many days in the last 3 months did you miss work or school because of your headaches?	0 (0–3)	0 (0–3)	0 (0–2)	
How many days in the last 3 months was your productivity at work or school reduced by half or more because of your headaches? (Do not include days you counted in question 1 where you missed work or school.)	3 (1–5)	3 (1–6)	0 (0–5)	0.1 ^#^
On how many days in the last 3 months did you not do household work (such as housework, home repairs, and maintenance, shopping, caring for children and relatives) because of headaches?	4 (2–6)	4 (2–7)	3 (1–5)	0.1 ^#^
How many days in the last 3 months was your productivity in household work reduced by half or more because of headaches? (Do not include days you counted in question 3 where you did not do household work.)	4 (2–6)	4 (2–6)	3 (1–5)	0.1 ^#^
On how many days in the last 3 months did you miss family, social, or leisure activities because of your headaches?	3 (1–5)	3 (1–5)	2 (0–4)	0.01 *
MM: Menstrual migraine; NMM: Non-menstrual migraine.				
a—Mood’s test with exact calculation				

* Significant, ^#^ Not Significant.

## Data Availability

The datasets used and/or analyzed during the current study are available from the corresponding author upon reasonable request and will be provided by masking the identification of the individuals.

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
