# Peer review of "Study to Determine the Prevalence of Menstrual Migraine in Reproductive-Age Women in Saudi Arabia"

_healthcare, 2024, doi:10.3390/healthcare12030317_

Round 1
Reviewer 1 Report
Comments and Suggestions for Authors
This is study is a cross-sectional study in Saudi Arabia involving 2130 women revealed that 18.6% had migraines, with 57.9% of them experiencing menstrual migraines, and identified a correlation between higher BMI and increased incidence of menstrual migraines, highlighting the need for effective management strategies to improve the quality of life for women with premenstrual syndrome and menstrual migraines. While grappling with some limitations, it shines in its expansive participant pool, and thorough data collection. The findings suggest a need further research, pointing toward a deeper exploration of risk factors and management strategies that could significantly enhance the well-being of those grappling with menstrual migraines.
Strengths:
1. With 2130 participants, this study’s large sample size, ensures a robust dataset for analysis, allowing for enhanced application of the findings to the broader Saudi Arabian population.
2. The study collects extensive data on various aspects, including migraine prevalence, severity, duration, associated symptoms, and the impact on daily activities. This comprehensive approach contributes to a nuanced understanding of menstrual migraines and their implications.
3. The study identifies key risk factors associated with menstrual migraines, such as BMI, family history, and trigger factors like sleep disturbances. This identification serves as a valuable guide for future research endeavors and can inform clinical practices for more effective management.
Weaknesses
1. Some of the data could have been represented graphically as to enhance the ease of readability and visualization of the article
2. Discussion way too long; the discussion reads like a repetition of the results section. The discussion should also include discussion in relation to possible causes and literature based correlation and explanation for the results observed.
3. The study's reliance on cross-sectional data restricts its ability to establish causal relationships or observe changes over time. Longitudinal studies would offer a more dynamic perspective on the evolution and impact of menstrual migraines.
4. Skimming the Surface of Comorbidities: While briefly acknowledging the use of antipsychotic drugs, the study doesn't delve into potential comorbid psychiatric conditions. A deeper exploration of mental health aspects and their interplay with menstrual migraines could enrich the study's depth.
5. A deeper explanation of the Data analysis methods
6. Despite linking migraine attacks to the menstrual cycle, the study lacks detailed information on cycle patterns, such as irregularities or hormonal fluctuations.
7. Depending on participants to recall and report their symptoms introduces a potential bias.Incorporating tools like migraine diaries or clinical assessments could enhance the accuracy and reliability of the data.
Additional grammatical errors and suggested corrections
1. Aim 3 Original lines126.127 3. To identify potential aggravating for developing menstrual migraines in women with PMS
Correction: To identify potential factors aggravating the development of MM in women with PMS.
2. Why is migraine capitalized in the discussion? And If you are going to abbreviate, keep with the abbreviations throughout the article. Example: At time you say menstrual migraine, then at time you put MM. Be consistent throughout the article.
3. Original lines 267-261: HIT-6 score and Migraine severity were not significant, the duration of migraine attacks was almost equal in both the groups of women with menstrual Migraine and non-menstrual Migraine, while MIDAS scores were higher among the self-reported Menstrual Migraine compared to the non-menstrual Migraineurs
Correction: HIT-6 score and migraine severity were not significant. The duration of migraine attacks was almost equal in both groups of women with MM and non-MM, while MIDAS scores were higher among self-reported MM sufferers compared to non-menstrual migraineurs.
4. Original line 264: This study observed that a total of 2130, most participants 1733(81.4%) suffered from PMDD without migraines, nearly 301 (14.1%) were suffering from migraine with Premenstrual symptoms, and very minimal participants 96 (4.5%) had migraines without PMDD.
Correction: This study observed that out of a total of 2130 participants, most specifically, 1733(81.4%), suffered from PMDD without migraines. Nearly 301 (14.1%) were experiencing migraines with premenstrual symptoms, and a minimal number of participants (4.5%) had migraines without PMDD.
5. Original line 266-267: The average age of the participants was 28 years with an interquartile range of 22-40, among the Premenstrual migraineurs were 33 years with a 24-40 years interquartile range, whereas non-MM Migraine is experienced in the early ages 24 years (21–35 years).
Correction: The average age of the participants was 28 years with an interquartile range of 22-40. Among the premenstrual migraineurs, the average age was 33 years with an interquartile range of 24-40, whereas non-MM is experienced at an early age of 24 years (21–35 years).
6. Original lines 324-331: Women with MM in this study stated nearly 8(5-12) attacks this year with the menstrual cycle. around 73% of them reported moderate to severe pain, nausea 68%, photophobia 86.1%. a higher impact of migraine in terms of higher HIT-scores 66(63-69), more migraine days, and longer duration of attacks, PMS symptoms were similarly common in women with and without MM. nearly more than half of the menstrual migraineurs were very often they were unable to deliver their protein activities like personal home activities, and academic and occupational activities, and their productivity reduced to half in the last 3 months in nearly 4 (2-6) days.
Correction: Women with MM in this study reported nearly 8 attacks (with a range of 5-12) this year associated with the menstrual cycle. Around 73% of them reported moderate to severe pain, 68% experienced nausea, and 86.1% reported photophobia. Migraine had a higher impact in terms of elevated HIT-scores (66 with a range of 63-69), more migraine days, and longer attack durations. PMS symptoms were similarly common in women with and without MM.
Additionally, more than half of the menstrual migraineurs often found themselves unable to perform routine activities at home and experienced a reduction in productivity by half in the last 3 months, lasting nearly 4 days (with a range of 2-6).
In addition, I am not certain what “protein activities” you were referencing.

Reviewer 2 Report
Comments and Suggestions for Authors
Overall this paper was clear and easy to understand. Because the readership is international, I would be more specific in the objectives about regionality given that environmental factors are a contributor to migraines and therefore cannot be generalized.
Objective 1: To estimate the prevalence of menstrual migraines in reproductive women in Saudi Arabia. (as an example)
The author states several times throughout the paper that individuals had to have "regular" menstrual cycles, which should be defined.
The demographic eligibility criteria should be made clearer. The groupings of the migraines were clear but the population was not.
Overall, interesting paper.
Reviewer 3 Report
Comments and Suggestions for Authors
This study aimed to investigate the prevalence of premenstrual migraine in Saudi Arabia.
I have several concerns:
1: The introduction is too long, yet lacks logical flow. The authors failed to justify the aim of the study by not describing the magnitude of the problem, previous research, and gaps in knowledge. This section needs to be rewritten.
2: From revising the citations of the first paragraph, it is obvious that these references are unrelated to the text, raising serious concerns about the validity of the entire writing process. For example, the authors claimed prevalence rates of 20% and 10% among (both men and women of reproductive age); however, reference 1 provided the prevalence of migraine in US men and women >12 years by their SES and reached different conclusions. Again, reference 4 gives a different prevalence rate for migraine than that claimed by the authors.
3: The authors cited several non-peer-reviewed websites to get information that could be reached by conducting a good literature review.
4: Most previous national and international work was not cited.
5: Although the main goal of this study was to detect prevalence, no randomization was conducted, suggesting that the findings are not generalizable.
6: The authors claimed to use a valid questionnaire, yet no validity nor reliability approaches were applied. It is also incorrect to just translate a questionnaire, written in a different language and conducted on a different population, and directly use it without a validation study.
7: The details of the questionnaire should have been described in more detail.
8: No data collection procedures were mentioned.
9: The authors said that they excluded participants with certain medical conditions, yet they did not show how these medical conditions were determined.
10: The authors said that they (recruited women who had regular menstrual cycles with migraine headaches and were diagnosed with PMS and menstrual migraine). This suggests that the entire methodology is illegitimate. How can the authors confine the study population to women with migraine headaches and assess the prevalence of migraine headaches? How could someone assess the prevalence of diabetes, for example, among diabetic patients?!
11: The results only provided data about univariate analyses. Given, the multi-faceted nature of PMS and migraine headaches, multivariable regression analyses are mandatory.
12: The fact that those without PMDD had zero symptoms of PMT raises another concern about how PMDD was diagnosed. Were women with any PMT symptoms considered to have PMDD?! What is the definition of PMDD in the current study?
13: Several variables, especially dietary factors, PA, and other lifestyle factors, that are strictly associated with migraine headaches were not collected.
14: Given the invalid results, the discussion section becomes meaningless. Nevertheless, it is obvious that the authors just focused on comparing their results with previous studies rather than giving meaningful explanations.
15: Several limitations should be included in the limitation section.
16: No originality was provided.
Comments on the Quality of English Language
Extensive English language editing is needed.
Reviewer 4 Report
Comments and Suggestions for Authors
Authors present an interesting study aiming to evaluate the prevalence of menstrual migraine in a Saudi Arabia population. Unfortunately, I believe that there are several structural errors in the data collection, and authors should consider making major changes before considering resubmission.
Major concerns:
- What is the difference between the authors described "PRE menstrual migraine" and the internationally approved classification term "PURE menstrual migraine" (see ICHD-3 classification)? I strongly believe that authors should refer exclusively to the second one and put the former apart.
- line 165 "those who self-reported migraines" meaning subjects did not receive a migraine diagnosis from a neurologist in the past and there may be two biases: people who do not have migraine may self diagnose it; people who have migraine but are not aware of it may not participate to the study. All women of reproductive age should have been included and have completed the questionnaire, not just the ones self reporting migraine.
Minor concerns:
- line 160: according to the ICHD-3 for research purposes the diagnosis of pure menstrual migraine or menstrual related migraine should rely on prospective diaries, so authors' different choice should be listed among the limitations of the study.
- line 138: the validated questionnaire should be provided as well as its references for validation.
-line 217 "397" should not be between brackets, but, more important, they suffered from migraine in their self opinion or beause they met ICHD3 criteria?
- there is no mention of MIDAS and HIT6 results in the results section, a reader could just see them from the tables, but they should be better presented in the main text and the statistical analysis explained.
- the limitation of just looking at literate women excluding illiterate ones should be added.
Reviewer 5 Report
Comments and Suggestions for Authors
1. In the abstract, include the objective related to Saudi Arabia.
2. In the objectives before methodology, write that you only specifically analyzed data from Saudi Arabia.
3. Methods
a) English translation, and in Saudi Arabia, a questionnaire should be provided.
b) If the questionnaire was validated, the authors should provide the clinometrics of the study. If the questionnaire was based on previous studies, please cite them at the end of the question.
c) Provide photos of the questionnaire and the consent term requested for the participants as non-published supplementary material.
d) How much time was needed to answer the questionnaire?
4. Describe the power of the study in the manuscript
5. Please, review statistics. In the objective of the study, the authors wrote, “2. To evaluate the severity and frequency of menstrual migraines in women with PMS. 3. To identify potential aggravating for developing menstrual migraines in women with PMS.” However, to analyze this, the authors should do a comparative analysis, which was not done in their study. Please, provide a formal statistical analysis evaluation.
a) Include how the variables were distributed.
b) How were assessed confounding variables
6. A validation question. How did the authors assume that women were replying to the questionnaire? How did the authors ensure that a double fulfillment of the questionnaire by the same person did not occur?
Round 2
Reviewer 3 Report
Comments and Suggestions for Authors
Most of my comments were not delivered. I still believe that the methods and statistical analyses are not legit making the entire manuscript hard to be accepted.
Comments on the Quality of English LanguageNA
Reviewer 4 Report
Comments and Suggestions for Authors
Authors addressed almost every point, but still there is mention of pre-menstrual migraine see line 50-51, with the same abbreviation of pure menstrual migraine "Pre-menstrual migraine (PMM) coincides with the menstrual cycle in 50% of women with migraines, often being frequent and severe".
When referring to migraine, I believe that is more elegant to write using the singular (e.g., patient affected by migraine, not by migraines). If authors wanted to use the plural maybe they should be referring to migraine attacks instead.
Reviewer 5 Report
Comments and Suggestions for Authors
1. Please remove the numeration in the objectives. The structure should be a paragraph; remove “:”
2. The questionnaire should be included. The present format does not appear scientific. Also, if the questionnaire was based on previous data from the literature, the authors should cite the articles from where the questions originated. If this was done, clinometric properties should be provided.
3. Please remove the numeration from the manuscript. Numerations should be used for chapters and subchapters.
4. Still needs a formal statistical evaluation. Please provide a certificate as a non-published material by a statistician.
